

# Defensive behaviors of the Oriental armyworm *Mythimna separata* in response to different parasitoid species (Hymenoptera: Braconidae)

Jincheng Zhou, Ling Meng and Baoping Li

School of Plant Protection, Nanjing Agricultural University, Nanjing, Jiangsu, China

## ABSTRACT

This study examined defensive behaviors of *Mythimna separata* (Lepidoptera: Noctuidae) larvae varying in body size in response to two parasitoids varying in oviposition behavior; *Microplitis mediator* females sting the host with the ovipositor after climbing onto it while *Meteorus pulchricornis* females make the sting by standing at a close distance from the host. *Mythimna separata* larvae exhibited evasive (escaping and dropping) and aggressive (thrashing) behaviors to defend themselves against parasitoids *M. mediator* and *M. pulchricornis*. Escaping and dropping did not change in probability with host body size or parasitoid species. Thrashing did not vary in frequency with host body size, yet performed more frequently in response to *M. mediator* than to *M. pulchricornis*. Parasitoid handling time and stinging likelihood varied depending not only on host body size but also on parasitoid species. Parasitoid handling time increased with host thrashing frequency, similar in slope for both parasitoids yet on a higher intercept for *M. mediator* than for *M. pulchricornis*. Handling time decreased with host size for *M. pulchricornis* but not for *M. mediator*. The likelihood of realizing an ovipositor sting decreased with thrashing frequency of both small and large hosts for *M. pulchricornis*, while this was true only for large hosts for *M. mediator*. Our results suggest that the thrashing behavior of *M. separata* larvae has a defensive effect on parasitism, depending on host body size and parasitoid species with different oviposition behaviors.

## INTRODUCTION

Since successful parasitism by parasitoids results in the death of the host, natural selection should favor the evolution of host defenses against parasitoids. Herbivore insects perform a wide variety of defense mechanisms in response to their parasitoids, including morphological, chemical, physiological, and behavioral traits (*Gross, 1993*; *Godfray, 1994*). Behavioral defenses are observed in a number of herbivorous insect groups, conferring protection against attacking parasitoids (*Gross, 1993*). For example, lepidopteran larvae perform a wide array of behavioral defenses in response to parasitoids, and these defenses fall into three broad categories: evasive, aggressive, and associative behaviors (*Gentry & Dyer, 2002*; *Greeney, Dyer & Smilanich, 2012*). Yet these categories of defensive behaviors

Corresponding author
Baoping Li, lbp@njau.edu.cn

include many variations, combinations, and modifications of innumerable potential life history strategies and behaviors, and their expression or employment may vary ontogenetically, temporarily, or in response to different enemies (*Stamp, 1982*; *Cornell, Stamp & Bowers, 1987*; *Allen, 1990*; *Greeney, Dyer & Smilanich, 2012*). Many studies have shown that host larval resistance to attacking parasitoids increases with age (and therefore body size) (*Gross, 1993*; *Firlej et al., 2010*; *Kageyama & Sugiura, 2016*). Such resistance can be achieved by a combination of morphology (e.g., increasingly tough exoskeleton) and behaviors associated with host age (or size) (*Brodeur, Geervliet & Vet, 1998*; *Yazdani, Glatz & Keller, 2015*; *Ameri, Rasekh & Michaud, 2014*; *Kageyama & Sugiura, 2016*). These defensive mechanisms often increase host handling time, shape parasitoid host-preference (*Lucas, Coderre & Brodeur, 1997*; *Potting, Vermeulen & Conlong, 1999*) and likely decrease parasitism success by parasitoids (*Gross, 1993*).

Individual herbivorous insect species are often attacked by more than one species of parasitoids (*Hawkins, 1984*), which is referred to as parasitoid species loads (*Godfray, 1994*). If a host species is parasitized by different parasitoid species that use different attacking tactics, selection can favor variation in host defenses that encode this information. According to this theory, we hypothesize that a larval host should behave differently in defense against different parasitoid species that have dissimilar oviposition behaviors. We tested this hypothesis using the oriental armyworm, *Mythimna separata* (Walker) as the host attacked by two larval parasitoid species, *Microplitis mediator* (Haliday) and *Meteorus pulchricornis* (Wesmael). The host larvae exhibit active behaviors in defense against parasitoids (*Lauro et al., 2005*; *Chu et al., 2014*). Its two parasitoid species differ in approaching the host: *M. mediator* females climb on the host to sting it with their ovipositors (*Wang et al., 1984*; *Arthur & Mason, 1986*), while *M. pulchricornis* females stand at a close distance from the host to make the stinging (*Yamamoto, Chau & Maeto, 2009*).

The host *M. separata* is a polyphagous pest of grain crops, causing major losses in crop production annually in China and other Asian countries (*Sharma, Sullivan & Bhatnagar, 2002*; *Jiang et al., 2014*). Each year it migrates by a seasonal, multi-generation, long-distance roundtrip between southern and northern China (*Jiang et al., 2011*). *Microplitis mediator* is known to attack the host species in the family Noctuidae and Geometridae (Lepidoptera). It was one of the dominant parasitoids of *M. separata* lavae in China (*Wang et al., 1984*; *Arthur & Mason, 1986*; *He, 2004*; *Li et al., 2006*). *Meterous pulchricornis* is a generalist parasitoid that is known to attack hosts in at least 12 families of the Lepidoptera (*Maeto, 1990*; *Harvey, Sano & Tanaka, 2010*; *Malcicka & Harvey, 2014*; *Xu et al., 2016*). Both parasitoid species have a similar preference for host stages (*Foerster & Doetzer, 2003*; *Lauro et al., 2005*; *Li et al., 2006*; *Liu & Li, 2006*). *M. pulchricornis* is approximately twice the size of *M. mediator* (*Malcicka & Harvey, 2014*). Both parasitoids are important biological control agents of various noctuid pests and have been introduced widely to control both natural and novel hosts (e.g., *Arthur & Mason, 1986*; *Fuester et al., 1993*; *Berry, 1997*; *Berry & Walker, 2004*; *Li et al., 2006b*; *Liu & Li, 2006*; *Chhagan, Stephens & Charles, 2008*; *Li et al., 2010*).

To test our hypothesis, we first determined how defensive behaviors of *M. separata* larvae varied with their own body size or two parasitoid species, and then examined how

host defensive behavior and body size influenced the likelihood of stinging and handling time of the two parasitoids. The understanding of the effectiveness of behavioral defenses sheds light on the evolution of host-parasitoid behavioral interactions, and helps to explain why this host species is more often parasitized by one parasitoid than by the other.

## MATERIALS AND METHODS

### Insects preparations

Oriental armyworms, *M. separata*, were provided by the Research Institute of Agriculture and Forestry of Hebei province, China, in 2013 and since then has been maintained in the insectary. Larvae were reared on semi-artificial diets (*Bi, 1989*). They were reared in groups of 40–60 from neonates in glass jars (9 cm height and 20 cm diameter). Pupae were collected in a plastic box (5 cm height and 8 cm diameter) for adult emergence. Adults in a group of 40–50 were placed in a rectangular cage, where a 10% honey solution was provided as food via a large cotton ball and pieces of nylon rope were suspended from the cage roof as substrates for egg deposition. Eggs were collected in a petri-dish with a soft brush and maintained for larval hatching. The 2nd or 3rd instar larvae, weighted from 1.5 to 50 mg, were used as hosts in the experiment.

The parasitoid *M. mediator* was provided by the Research Institute of Agriculture and Forestry of Hebei province, China, in 2013 and since then has been maintained with *M. separata* 2nd or 3rd instar larvae as hosts in the insectary. Host larvae in groups were placed in a plastic box and then two female wasps were released. After 24 h the wasps were removed and the larvae were reared on the semi-artificial diets until offspring parasitoid larvae egressed from host larvae and pupated. Parasitoid pupae were collected in groups in a glass tube (8 cm height ×2 cm diameter) to allow adults to emerge. Emerged wasps were maintained in groups in a vial to allow mating for two days, during which time a 10% honey solution was provided as supplementary food via a piece of cotton thread. Four days old, naive female wasps were used in the experiment.

The parasitoid *M. pulchricornis* is thelytokous and its laboratory stock was established from rearing the tobacco cutworm *Spodoptera litura* larvae collected from soybean fields in the northern suburb of Nanjing in 2013. Thereafter it has been maintained using *M. separata* larvae as hosts. The 2nd or 3rd instar larvae were exposed in groups to parasitism in vials. The parasitoid pupae were collected in vials to allow adults to emerge. The emerged wasps were kept in glass tubes and provided with a 10% honey solution via a piece of cotton thread. Four to six days old, naive adults were used in the experiment.

### Experimental protocol

Host and parasitoid behaviors were observed on potted wheat seedlings in a transparent cage (12 cm height and 4.5 cm diameter, the top being covered by nylon gauze). *Triticum aestivum* L wheat seeds were planted in pots (9 cm height and 6 cm in diameter) with peat moss-sand and soil. Wheat seedlings grown to a height of 10 cm with seven leaves were used as food plants for host larvae. A larva was placed on potted wheat seedlings after being weighed (Mettler Toledo AL204-IC Electronic Microbalance, accurate to 0.0001 g; Mettler Toledo, Columbus, OH, USA), and an hour later a female wasp was released into the cage.

**Table 1  Description of parasitoid and host behaviors recorded.**

|  | Behavior | Description | Measurement |
|---|---|---|---|
| Host | Thrashing | Raising and quickly shaking the head. | Count |
|  | Escaping | Moving by quickly crawling away from the approaching wasp by more than three times its own body length on the plant | Binary |
|  | Dropping | Falling off the plant either to the ground or by hanging on a silken thread. | Binary |
| Wasp | Stinging | Inserting the ovipositor inside the host body. | Binary[*] |
|  | Host-handling time | The interval between first contact with the host and completion of ovipositor stinging. | Continuous |

Notes.

[*]The ovipositor stinging by *M. pulchricornis* was defined as being successful when the insertion lasted more than one second and a characteristic wing-flapping occurred when withdrawing the ovipositor from the host (*Zhang, Li & Meng, 2014*). For *M mediator*, a bout of stinging always results in an egg deposition (*Wang et al., 1984*).

The host and wasp were observed continuously for a maximum of 30 min, during which time the frequency of behaviors exhibited by the host and wasp (Table 1) were noted. The observation was terminated before 30 min if either the wasp made a successful sting of the host with its ovipositor, or the host fell off the plant or escaped before a ovipositor sting was made. Each parasitoid female was tested once. A total of 86 females (replicates) for each parasitoid species were observed.

## Statistical analysis

We first analyzed host defensive behaviors as a function of host body weight and parasitoid species, by applying a logistic model (link = logit) to estimate the probability of host dropping and escaping, and by a simple linear model to estimate frequency of host thrashing (weighted by observation time). We then used a logistic model to estimate the probability of ovipositor stinging, and a generalized linear model with gamma distribution to evaluate host handling time, which is conveniently described by gamma distribution (*Kohlmann, Matis & Risenhoover, 1999*), Host thrashing, host body size, and parasitoid species were tested as predictor variables. Overdispersion was taken into account where appropriate via empirical estimation of scaling parameters (*Faraway, 2016*). We did not consider host dropping and escaping behaviors in estimating parasitoid handling time and stinging because the behavioral observation was designed to be terminated when the host dropped from the plant or escaped. Robust standard errors were used to control for the violation of the distribution assumption for the parameter estimation (*Croux, Dhaene & Hoorelbeke, 2004*). Analyses were carried out with R ver. 3.3.1 (*R Core Team, 2016*).

## RESULTS

The probability of host escaping was not significantly influenced by host body weight ($\chi^2 = 3.37$, $P = 0.06$), wasp species ($\chi^2 = 1.59$, $P = 0.21$), or their interaction ($\chi^2 = 1.33$, $P = 0.25$). The proportion of hosts escapeding was 31.4% (standard deviation, SD = 46.7%, $n = 86$) in response to *M. mediator* (Fig. 1A) and 40.7% (SD = 49.2%, $n = 86$) to *M. pulchricornis* (Fig. 1B).

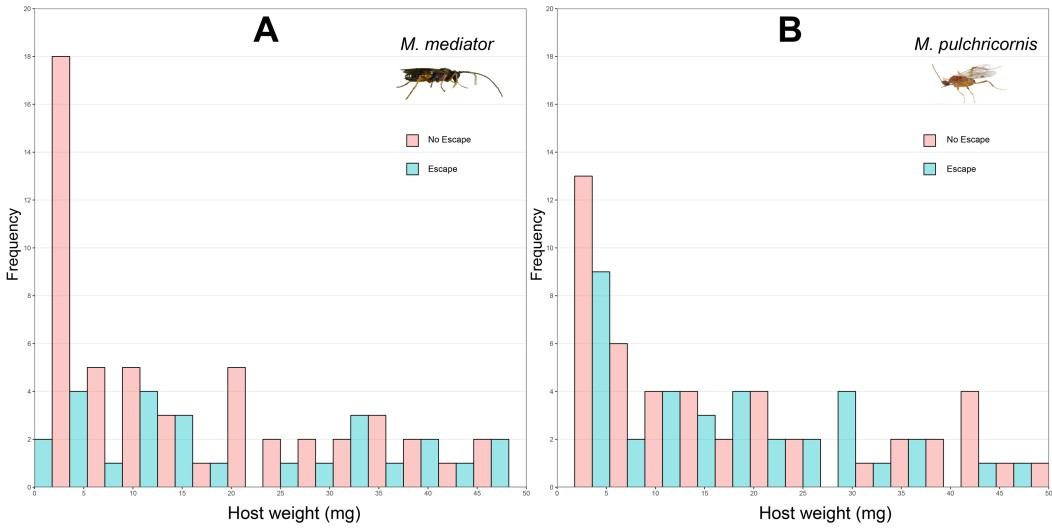

**Figure 1** Distribution of body weight for *Mythimna separata* larvae with or without escaping in response to *Microplitis mediator* (A) and *Meteorus pulchricornis* (B).

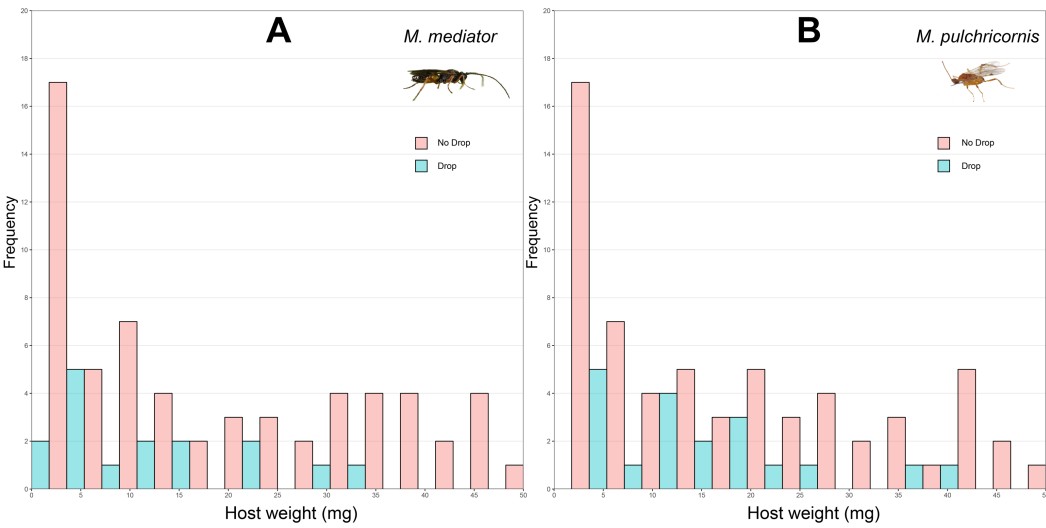

**Figure 2** Distribution of body weight for *Mythimna separata* larvae with or without dropping in response to *Microplitis mediator* (A) and *Meteorus pulchricornis* (B).

The probability of host dropping was not significantly influenced by host weight ($\chi^2 = 2.80$, $P = 0.09$), wasp species ($\chi^2 = 0.36$, $P = 0.55$), or their interaction ($\chi^2 = 0.38$, $P = 0.54$). The proportion of hosts droppeding was 18.6% (SD = 39.1%, $n = 86$) in response to *M. mediator* (Fig. 2A) and 22.1% (SD = 41.7%, $n = 86$) to *M. pulchricornis* (Fig. 2B).

Host thrashing frequency was not significantly affected by the interaction between host body weight and wasp species ($F_{1,166} = 1.74$, $P = 0.19$), or host weight ($F_{1,166} = 1.95$, $P = 0.16$), but it was influenced by wasp species ($F_{1,166} = 5.59$, $P < 0.05$). Thrashing

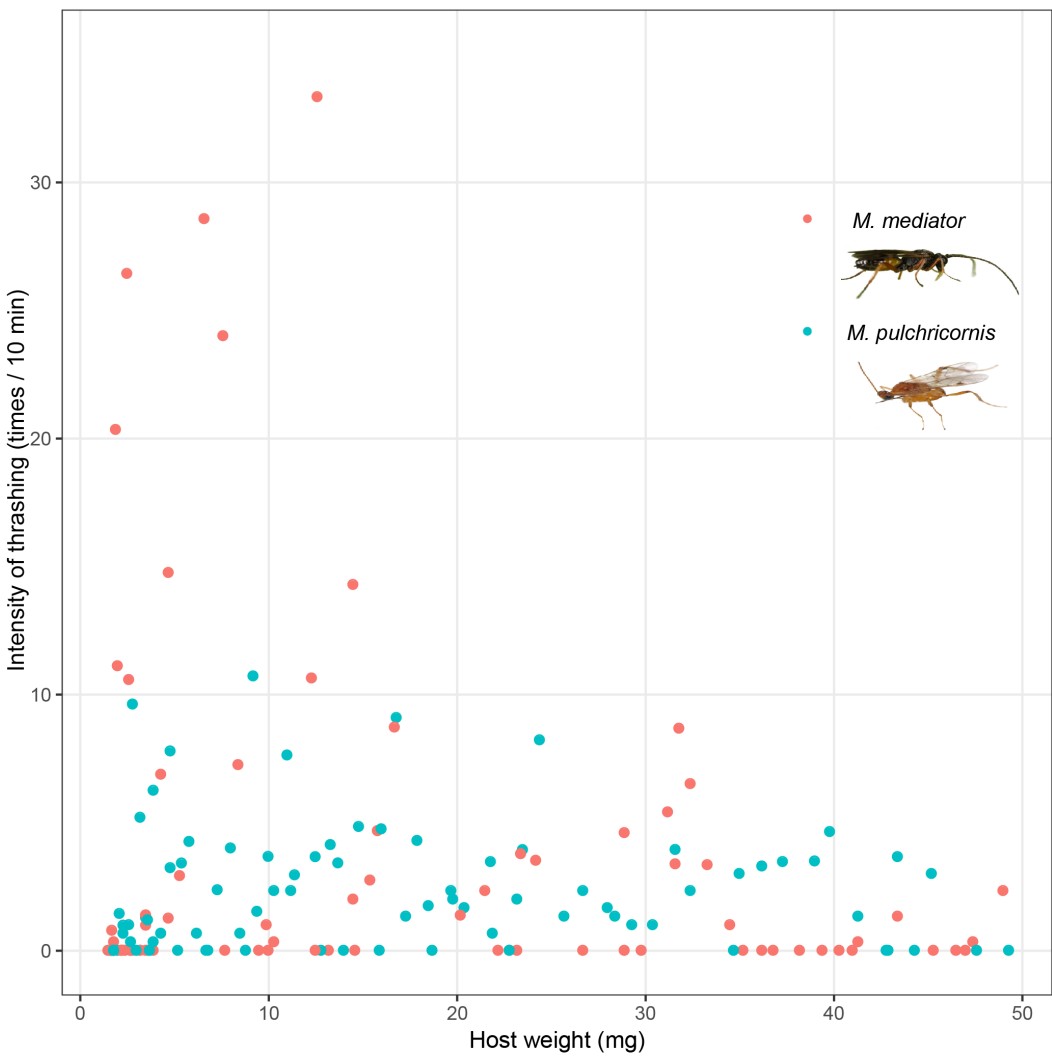

**Figure 3** **Intensity of thrashing of *Mythimna separata* larvae in response to *Microplitis mediator* and *Meteorus pulchricornis*.**

frequency was nearly 10 times higher in response to *M. mediator* (on average 21.11 times/10 min) than in response to *M. pulchricornis* (on average 2.23 times/10 min) (Fig. 3).

The probability of the stinging by parasitoids was affected by a 3-way interaction among host body weight, thrashing frequency and parasitoid species ($\chi^2 = 4.11$, $P < 0.05$). For *M. mediator*, it decreased with thrashing frequency when the host was larger but increased with it when the host was smaller (Fig. 4A). For *M. pulchricornis*, however, it decreased with thrashing frequency regardless of host body weight (Fig. 4B).

Host-handling time was influenced by a 2-way interaction between host body weight and parasitoid species ($\chi^2 = 7.17$, $P < 0.01$). It decreased with host body weight for *M. pulchricornis*, but not for *M. mediator* (Fig. 5A). It increased with host thrashing frequency

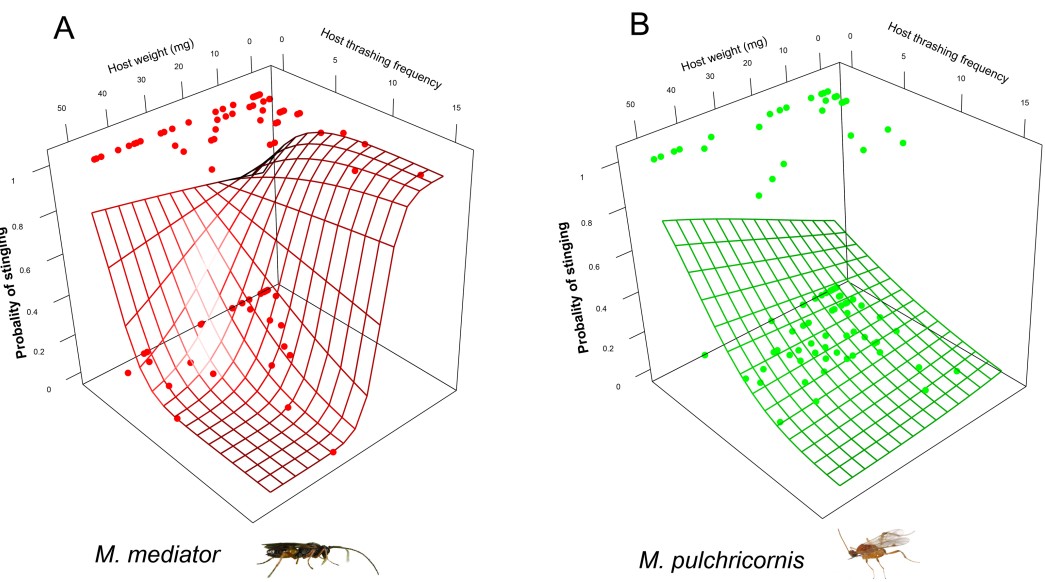

**Figure 4** **Probability of stinging as a function of both *Mythimna separata* larval body weight and thrashing frequency for *Microplitis mediator* (A) and *Meteorus pulchricornis* (B).** The grid surface is predicted probabilities from logistic regression models.

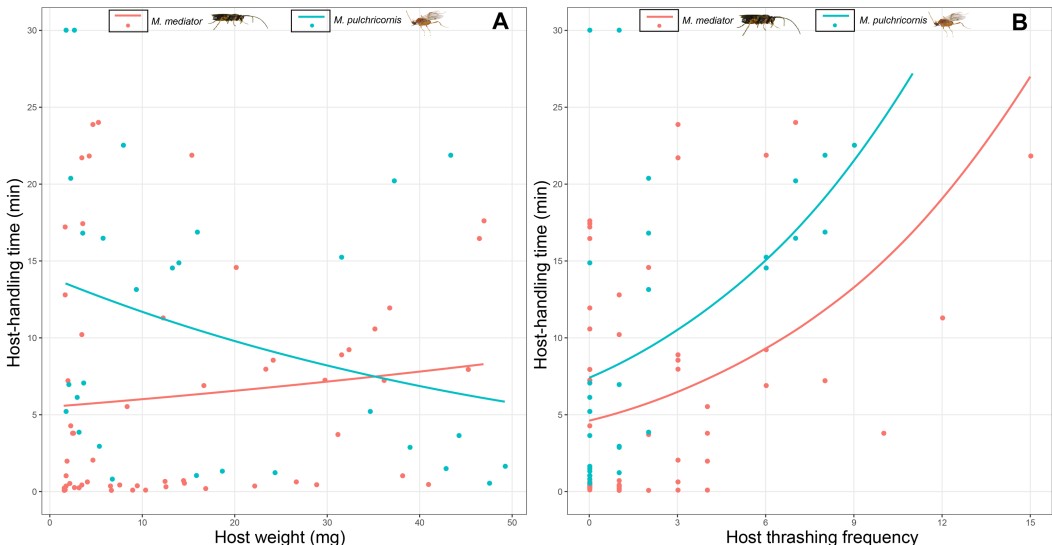

**Figure 5** **Host-handling time as a function of *Mythimna separata* larval body weight (A) and thrashing frequency (B) for *Microplitis mediator* and *Meteorus pulchricornis*.**

in both parasitoid species ($\chi^2 = 13.81$, $P < 0.001$), yet was overall shorter by 36% for *M. mediator* than for *M. pulchricornis* (Fig. 5B).

## DISCUSSION

Our data support the hypothesis that *M. separata* larvae adjust their defensive behaviors in response to different species of parasitoids that differ in oviposition behaviors. We

showed that the larvae, whether small or large in size, thrashed the head more frequently in response to *M. mediator* than to *M. pulchricornis*, though they dropped and escaped in the same likelihood in response to them. As a result of such thrashing defenses, *M. mediator* spent shorter time to handle the host than did *M. pulchricornis*. Furthermore, *M. mediator* females increased the likelihood of stinging when smaller larvae thrashed more frequently, while *M. pulchricornis* females decrease it when confronting such larvae.

The different oviposition behaviors between these two parasitoid species can explain the variable behavioral responses of *M. separata* larvae to them and the consequences of these defenses on them. *Microplitis mediator* females have to climb onto the host body to insert the ovipositor into the host (*Wang et al., 1984*; *Arthur & Mason, 1986*), probably owing to its shorter ovipositor (*He, 2004*). *Meterous pulchricornis* females, however, with a longer ovipositor, furtively approach the host and make an attack while maintaining a distance from it (*Yamamoto, Chau & Maeto, 2009*). It is expected that climbing onto the host body by *M. mediator* females would incur more frequent thrashing than maintaining a distance from the host by *M. pulchricornis* females. However, *M. mediator* females spent shorter time to handle the host than did *M. pulchricornis* females, which suggests that the host defensive behavior is more effective against *M. pulchricornis* than against *M. mediator*. Smaller or younger hosts are often less defensive than larger or older hosts, owing mainly to stronger thrashing behaviors or tough exoskeleton in larger or older hosts (*Brodeur, Geervliet & Vet, 1998*; *Firlej et al., 2010*; *Ameri, Rasekh & Michaud, 2014*; *Yazdani, Glatz & Keller, 2015*; *Kageyama & Sugiura, 2016*). Thrashing can be a powerful counterattack to avoid contact with ovipositors, and may even result in dislodging or injuring parasitoids in some larvae (*Myers & Smith, 1978*; *Stamp, 1982*; *Heinz & Parrella, 1989*). On the other hand, remaining motionless can also be an effective defense against parasitoids (*Richerson & Deloach, 1972*; *Rotheray, 1981*), for in some parasitoid species visual cues from moving larvae are necessary for a successful oviposition (*Nakamatsu & Tanaka, 2005*; *Yamamoto, Chau & Maeto, 2009*). Our observation that *M. pulchricornis* spent longer host-handling time and had an overall lower likelihood of stinging than did *M. mediator* The suggestion that *M. separata* is more effective to defense against attacks by *M. pulchricornis* than by *M. mediator* may partly explain why *M. separata* is less often parasitized by *M. pulchricornis* (*Jiang et al., 2011*) than by *M. mediator* (*Li et al., 2006b*; *Luo et al., 2013*).

Our results suggest that *M. separata* larvae do not adjust dropping and escaping behaviors in response to different parasitoid species, though escaping is more often exhibited than dropping. Dropping off the plant can be an effective defense against parasitoids for lepidopteran larvae (*Greeney, Dyer & Smilanich, 2012*) and occurs in many other insect taxa as well (*Gross, 1993*), probably because it removes the host and its associated chemical and sensory cues from the immediate vicinity of the parasitoid, making it difficult to relocate. However, dropping to the ground can be costly for phytophagous insects, as it reduces feeding time and increases mortality risk (*Greeney, Dyer & Smilanich, 2012*; *Gish & Inbar, 2006*). Larvae that have dropped to the ground often face predation from ground predators, such as ants, spiders and carabids (*Winder, 1990*; *Lövei & Sunderl, 1996*; *López & Potter, 2000*). As an alternative to dropping, another evasive behavior, escaping, was observed in *M. separata* larvae, whereby the caterpillar swiftly moved away on the leaf

once touched by an approaching parasitoid. By escaping, host larvae can avert the first attack by parasitoids, which often give up further attack (*Gross, 1993*). Many studies have shown that the propensity to escape or drop shifts during larval development, being higher in earlier (therefore, smaller in body size) than later (larger) instar hosts (*Stamp, 1982*; *Cornell, Stamp & Bowers, 1987*; *Allen, 1990*; *Lucas, Coderre & Brodeur, 1997*). We assume that the host stages (2nd and 3rd instar) we tested were not wide enough to show the effect.

In total, our findings have several broad implications. First, *M. separata* larvae can adjust their thrashing behaviors in response to different parasitoid species that have dissimilar oviposition behaviors. Second, the defensive behaviors of the host are more effective against *M. pulchricornis* than against *M. mediator*, which may help to explain the difference between the two parasitoid species in importance as biological control agents of *M. separata*.

## ACKNOWLEDGEMENTS

We would like to thank Li Jiancheng for providing the host and its parasitoid *M. mediator*, and John Lazarus for his suggestions of improving the language.

### Funding

This work was supported by the National Natural Science Foundation of China (NSFC-31570389). The funders had no role in study design, data collection and analysis, decision to publish, or preparation of the manuscript.

### Grant Disclosures

The following grant information was disclosed by the authors:
National Natural Science Foundation of China: NSFC-31570389.

### Competing Interests

The authors declare there are no competing interests.

### Author Contributions

- Jincheng Zhou conceived and designed the experiments, performed the experiments, analyzed the data, contributed reagents/materials/analysis tools, wrote the paper, prepared figures and/or tables, reviewed drafts of the paper.
- Ling Meng and Baoping Li conceived and designed the experiments, analyzed the data, contributed reagents/materials/analysis tools, wrote the paper, reviewed drafts of the paper.

### Data Availability

The raw data has been supplied as Data S1.

### Supplemental Information

Supplemental information for this article can be found online at http://dx.doi.org/10.7717/peerj.3690#supplemental-information.

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
