# Peer review of "Defensive behaviors of the Oriental armyworm Mythimna separata in response to different parasitoid species (Hymenoptera: Braconidae)"

_PeerJ, doi:10.7717/peerj.3690_

## Round 0.1 · original submission · Major Revisions

· Academic Editor

Major Revisions

Please revise your manuscript, taking care to address all of the concerns raised by the reviewers.

Reviewer 1 ·

Basic reporting

The presentation is sufficiently clear. The references are mostly quite old. Figure 1 has an error as A and B are the same figure but listed as representing the two different species. I found Fig. 4 difficult to interpret. What is the grid?

I find it impossible to interpret the importance of this study as there is no background on the relevance of these parasitoids to the natural situation for the larvae.

Li 142 need to add SD
Li 200 correct Myers and Smith from Myer and Simith

Experimental design

This seemed reasonable if the aim was simply to look for differences in the host responses to the two parasitoids without any attempt to understand the relevance to populations of Mythimna separata.

Need to include units for thrashing frequency in the text. These levels seem very low on a per minute basis. Is that really the relevant time unit to use?

Validity of the findings

The findings are valid but would be more interesting if put into a broader context even by giving an idea of the levels of parasitism in local field populations and further context for the study.

Additional comments

It isn't clear if this is part of a larger study on this system.

·

Basic reporting

The article describes the defensive behaviors of Mythimna separata larvae in response to two parasitoid species, Microplitis mediator and Meteorus pulchricornis that differ in the oviposition behavior. Authors use laboratory insects to make the behavioral observations. In addition, they include the host body size as a variable influencing the behaviors.
The English needs to be improved in some parts, I highlighted several instances when this was necessary (see General comments). In addition, the article will benefit in terms of clarity if the authors state clearly their hypothesis, or what they expect at the introduction. In general, the context is well presented, although a bit of more information on natural history of the system would be ideal (see General comments), as well as clearly stating what are the knowledge gaps that this article fills.
Table and figure captions need to be re-written in order to be self-explanatory, specially Fig. 4 (see General comments for more details). In all captions, authors must change “Mythimna separate” by “Mythimna separata”. Raw data is provided to the reviewer showing all the necessary info to re-run the analyses, except for the observation time. In the data sheet, it is necessary to change “confrontation.time” by handling.time

Experimental design

The article is an orignial contribution for researchers working on insect pests and host-parasitoid interactions. However, as mentioned above, the hypotheses and the knowledge gaps that this article fills are not clearly stated.
Data seems to be correctly analyzed (but see General comments). The results could be important when choosing parasitoid species for biological control.

Validity of the findings

I think that the discussion needs to be improved in order to show what the implication of the results are (both in terms of evolution and biological control), and the final paragraph needs more work to reorganize the ideas and leave a clear take home message.

Additional comments

It would be helpful to include the objectives in the Abstract, as well as the hypothesis.
L. 31. Delete “make the”
L. 38-39: re-write “similar in slope for both parasitoids yet on a higher intercept for M. mediator than for M pulchricornis” in biological terms instead of statistical ones

L. 40. Replace “realizing an ovipositor sting” by “stinging”

L.61: add “larval” before host, because eggs, pupae and adult insect do not grow as they aged

L.68: replace “using a” by “in the”, and replace “which was composed of” by “including”. In addition, it will be important to add in this paragraph if host and parasitoids are native and coexists frequently in nature in order for them to have chances to coevolve oviposition and defensive behaviors

L. 101. Replace “with” for “a”

L.119: replace the first “was” for “were”

L. 120: Remove “on the plant”

L. 129. It is necessary to explain why authors used a gamma distribution to model host handling time by parasitoids, instead of a linear model

L. 130. Add “and” at the beginning of the sentence

L. 133. Remove “on the plant”

L. 142. Add “SD=” before 49.2%

L. 144-6. In this paragraph authors are reporting the results of host dropping behavior, thus, the results of host escaping should be reported in the first paragraph

L.140-50. Re-write the sentence to “Thrashing frequency was almost 10 times higher in response to M. mediator (on average 2.11 [explain clearly what this number is, thrash/min?]) than in response to M. pulchricornis (on average 0.22 [explain clearly what this number is])”

L.151. Delete “realized”

L. 152: It seems that authors need to change “In response to M. mediator,” by “For M. mediator,” The same in line 154 for the other parasitoid species

L. 159. Please specify what variable is that P- value referring to.

L. 169-70 Data supporting this statement was not presented in the Results. Either include it (I think is the best) or add here that is unpublished data. In addition, it will be clearer if authors move this sentence after the following one.

L. 178-9: a reference or data from the authors is needed here

L.180: as it is written it seems that authors are referring to parasitoids when they say that “Studies of parasitoids suggest that the propensity to escape rather than defend”, although it should be related to host. Please, re-write

L. 184. It would be interesting to discuss something related to the fact that they found less dropping than escaping

L189. A reference, data or observations from the authors is needed after “probably owing to its shorter ovipositor” handling time

L. 189. Change Pulchricornis by pulchricornis

L. 192-6. It would be interesting to discuss the biological meanings of these results. For example, why more frequent thrashing by the host increased the likelihood of being attacked by M. mediator when it is supposed to be a defensive behaviour? In addition, it is not clear what authors are referring to when they say “the risk of attack”, is that the likelihood of being attacked? Is that is the case, authors need to use the same terms, and to discuss their results for handling time.

L. 197-204: this last paragraph needs order, as it is, it’s not clear what the message authors want to leave to the readers.

Figure 1. Change caption to “Distribution of body weight for Mythimna separata larvae escaping or not in response to Microplitis mediator (A) and Meteorus pulchricornis (B)”
In addition, it is necessary to separate the bars for each weight category in the figure as to make it clearer.

Figure 2. Same modifications as suggested in Fig. 1 for the caption and the figure.

Figure 4. Re-write the caption to include all variables in the model.

---

## Round 0.2 · accepted · Accept

· Academic Editor

Accept

Thank you for your careful attention to the revisions.